# An inventory of European data sources to support pharmacoepidemiologic research on neurodevelopmental outcomes in children following medication exposure in pregnancy: A contribution from the ConcePTION project

Joanne Given[1], Rebecca L. Bromley[2,3], Florence Coste[4], Sandra Lopez-Leon[5,6], Maria Loane[1]*

1 Faculty of Life and Health Sciences, Ulster University, Coleraine, Northern Ireland, United Kingdom, 2 Division of Neuroscience and Experimental Psychology, University of Manchester, Manchester, United Kingdom, 3 Royal Manchester Children's Hospital, Manchester Academic Health Science Centre, Manchester, United Kingdom, 4 Sanofi Aventis R&D, Chilly Mazarin, France, 5 Novartis Pharmaceuticals Corporation, One Health Plaza, East Hanover, NJ, United States of America, 6 Center for Pharmacoepidemiology and Treatment Science, Rutgers University, New Brunswick, NJ, United States of America

* ma.loane@ulster.ac.uk

## Abstract

### Background

Studies on medication safety in pregnancy are increasingly focusing on child neurodevelopmental outcomes. Establishing neurodevelopmental safety is complex due to the range of neurodevelopmental outcomes and the length of follow-up needed for accurate assessment. The aim of this study was to provide an inventory of European data sources for use in pharmacoepidemiologic studies investigating neurodevelopment following maternal medication exposure.

### Method

The EUROmediSAFE inventory of data sources in Europe for evaluating perinatal and long-term childhood risks associated with in-utero exposure to medication was updated by contacting colleagues across 31 European countries, literature review and internet searches. Included data sources must record at least one neurodevelopmental outcome and maternal medication use in pregnancy must be available, either in the data source itself or through linkage with another data source. Information on the domain of neurodevelopment, measure/scale used and the approach to measurement were recorded for each data source.

### Results

Ninety data sources were identified across 14 countries. The majority (63.3%) were created for health surveillance and research with the remaining serving administrative purposes (21.1% healthcare databases, 15.6% other administrative databases). Five domains of

**Data Availability Statement:** The EUROmediSAFE inventory of available data sources in 28 EU Member States for evaluating perinatal and long-term childhood risks associated with in-utero exposure to medication is available at http://www.euromedicat.eu/content/EUROmediSAFE Inventory_Finalv2_2018_07_06.pdf. The ConcePTION data source catalogue of population-based data sources that may be used for medication utilization and medication safety studies in pregnancy is available at https://www.imi-conception.eu/wp-content/uploads/2019/09/ConcePTION_D1.1_spreadsheet-containing-all-additional-data-sources-for-the-ConcePTION-Data-Source-Catalogue.pdf. All relevant data relating to 'An inventory of European data sources to support pharmacoepidemiologic research on neurodevelopmental outcomes in children following medication exposure in pregnancy' are within the manuscript and its Supporting Information files.

**Funding:** The ConcePTION project has received funding from the Innovative Medicines Initiative 2 Joint Undertaking under grant agreement No 821520 (authors JG, ML, FC and RB). This Joint Undertaking receives support from the European Union's Horizon 2020 research and innovation programme and EFPIA. https://www.imi.europa.eu/ The funders had no role in study design, data collection and analysis, decision to publish, or preparation of the manuscript.

**Competing interests:** I have read the journal's policy and the authors of this manuscript have the following competing interests: Florence Coste is a Sanofi employee. Sandra Lopez is a Novartis employee. This work was conducted following the ENCePP Code of Conduct http://www.encepp.eu/code_of_conduct/documents/ENCePPCodeofConduct.pdf

neurodevelopment were identified—infant development (36 data sources, 13 countries), child behaviour (27 data sources, 10 countries), cognition (29 data sources, 12 countries), educational achievement (20 data sources, 7 countries), and diagnostic codes for neurodevelopmental disorders (42 data sources, 11 countries). Thirty-nine data sources, in 12 countries, had information on more than one domain of neurodevelopment.

## Conclusion

This inventory is invaluable to future studies planning to investigate the neurodevelopmental impact of medication exposures during pregnancy. Caution must be used when combining varied approaches to neurodevelopment outcome measurement, the age of children in the data source, and the sensitivity and specificity of the outcome measure selected should be borne in mind.

## Introduction

Studies on drug utilisation in pregnancy report that up to 70–90% of women use one or more medications during pregnancy [1, 2]. Despite this, only 5% of all medications have been tested for use in pregnancy and appropriate safety information recorded on the medication patient information leaflet [3]. A review of drugs assessed by the Food and Drug Administration (FDA) reported that 97.7% of the drugs were classified as having an "undetermined" teratogenic risk in human pregnancy, and the mean time to determine a risk was 27 years [4]. There is therefore an urgent need for knowledge of medication use and safety during pregnancy.

Historically research has concentrated on congenital anomalies but there is increasing interest in the potential for medication exposure during pregnancy to adversely impact neurodevelopment (ND) [5, 6]. The term ND covers a diverse range of brain functions including intellectual abilities, language, attention, and cognition, but also encompasses motor development, social skills, behavioural and emotional regulation. Such diversity means that there are many outcomes which fall within the category of ND and even more numerous ways to define and measure functioning in these skill areas. As different cognitive, motor, and social skill sets mature at different ages certain effects will only become evident as age relevant skills emerge and mature. For example, the expected complexity of social skills as a two-year-old is far less than the complex abilities in both verbal and non-verbal social communication expected in the adolescent years. As different domains of ND may be differentially impacted upon by teratogen exposure a wide variety of outcomes must therefore be assessed at appropriate ages to establish ND safety [5].

It is a priority to increase efforts to detect medications which convey risk to the developing child's brain. This is a particular concern for medications, which affect the central nervous system and which can cross the placental barrier [7–12], such as the antiseizure medications (ASMs), antidepressants and antipsychotics. For example, exposure to the ASM valproate during pregnancy has been associated with reduced IQ scores, particularly verbal IQ, attention deficit hyperactivity disorder (ADHD) and Autism spectrum disorder (ASD) [13–15]. Isotretinoin exposure in utero has been found to reduce IQ scores, but had a more significant impact on visual-spatial skills [16]. Finally, there is conflicting evidence regarding the risk of ASD [17–21] following in utero exposure to selective serotonin reuptake inhibitor (SSRI) antidepressants. Recent investigations using detailed language assessments of every exposed child in the cohort raise the possibility that the primary deficit may be in the language domain and in

particular, pragmatic language [22]. There is therefore a clear risk of lifelong ND impairments associated with certain medication exposures and efforts should be made to detect those which carry this risk as soon as possible.

To date most evidence relating to the impact of medication exposure on ND has derived from observational studies and population-based cohort studies utilising electronic records. Both have inherent methodological limitations and strengths. Traditional observational cohort studies recruit pregnant women directly within hospital or community-based health care settings and the participants are followed up using study specific standardised protocols, often utilising direct blinded assessment of the child through the postnatal years, with good control over confounding variables. However, such methodologies may have lower statistical power, have relatively short follow-up periods (typically only up to pre-school age) and can be financially costly. Cohorts derived from population based electronic records alternatively, offer large numbers of exposed children often across a broader range of maternal indications. However, these are often based on diagnostic codes recorded or service referrals [23], data comes from multiple assessors who are not blinded to the medication exposure history of the child and often have more limited information on potential confounding variables (e.g. wider family history of disorders, parental intellectual level etc). Thus, pharmacoepidemiology research in relation to ND will require a combination of methodological approaches which cover a range of outcomes, with investigations extending into the adolescent years.

The growth of secondary data sources, with mother-baby linkages and large population sizes, raises the potential for timely evaluations of neurodevelopmental safety following maternal medication use during pregnancy. The aim of this study was to provide an inventory of European data sources with the potential to be used in pharmacoepidemiologic studies investigating ND in relation to maternal medication exposure. The objectives of the study were to capture how ND outcomes are recorded within these data sources and to consider their strengths and limitations for assessing ND outcomes.

## Material and methods

The Innovative Medicines Initiative ConcePTION Project is a large collaborative project between academic, regulatory and industry partners [24], with the primary aim to create a system of improved monitoring and communicating safety of medicines use in pregnancy and breastfeeding. One of the tasks of the ConcePTION project was to "identify data sources that can be used for medication utilisation and medication safety studies". An inventory of available data sources in 28 EU Member States for evaluating perinatal and long-term childhood risks associated with in-utero exposure to medication was published by the EUROmediSAFE consortium in 2018 [25]. The full EUROmediSAFE inventory is available at http://www. euromedicat.eu/content/EUROmediSAFE Inventory_Finalv2_2018_07_06.pdf. We reviewed, updated and extended the EUROmediSAFE inventory to provide the ConcePTION Consortium and other beneficiaries with a complete inventory of European data sources which could be considered for medication utilization and medication safety studies in pregnancy available at https://www.imi-conception.eu/wp-content/uploads/2019/09/ConcePTION_D1.1_ spreadsheet-containing-all-additional-data-sources-for-the-ConcePTION-Data-Source-Catalogue.pdf. This article relates specifically to the identification of data sources which could be used for investigations of longer-term ND outcomes in pharmacoepidemiologic studies.

### Identification of data sources

A number of different methods were used to identify potential sources of information. First, we contacted our colleagues in the EUROCAT (European surveillance of congenital

anomalies) network /EUROmediCAT (European congenital anomalies and medication safety) consortium with members in 21 countries and Euro-Peristat (European surveillance of perinatal health) with members in 31 countries. The purpose of the study was explained and they were invited to review the contents of the EUROmediSAFE inventory and to provide updates or add new electronic or linkable data sources that could potentially be useful for studies on medication use and safety in pregnancy in their country. They were asked to specifically consider data sources for capturing ND outcomes. This work was supplemented by a workshop held at a Euro-Peristat meeting in 2019 which was attended by approximately 50 Euro-Peristat members from across Europe and where we presented our findings and sought to identify additional sources in the countries that had not responded to our email requests.

Secondly, we conducted a literature review using the Embase database to identify data sources with outcomes following SSRI exposure during pregnancy in July 2019, which was updated in July 2020. SSRIs were used to identify data sources as the authors are conducting pharmacoepidemiologic studies on SSRI exposures in pregnancy, but other medications such as antiepileptic drugs could equally have been used. The search terms used are included in S1 File. The search was limited to the English language and publications within the last 10 years. Conference abstracts were excluded. Articles exploring ND outcomes were identified based on the title and the abstract and their sources checked against those already identified for the inventory.

The literature review was further supplemented by searches of national statistical organisation websites (for e.g. Statistics Denmark, Statistics Norway) and the https://www.birthcohorts.net/ website was searched to identify any missing birth cohorts with maternal medication exposure in pregnancy and ND outcomes [26].

### Eligibility for inclusion of data sources in this inventory

- Secondary source of electronic data with potential to be linked (i.e. primary purpose of collection was not for medication exposure investigations)

- Information on at least one ND outcome (e.g. behaviour, cognition, emotional regulation outcomes) using diagnostic codes, questionnaires, medical charts etc.

- Information regarding maternal medication use in pregnancy, either in the data source itself or through linkage with another data source.

### Exclusion criteria

Prospective Studies such as case reports, clinical studies, randomised controlled trials and adverse drug reaction databases were excluded as these have small sample sizes or selected populations.

### Classification of data sources

Data sources were classified according to the type of data available, see S2 File for more information:
Healthcare databases:

- Hospital (Admission/Episode/Discharge) databases

- Primary care databases

- Administrative health insurance claims databases

- Child surveillance databases

   Other administrative databases for the delivery of services, reimbursement of costs:

- Educational databases

- Register of disability

   Health surveillance and research databases:

- Disease registries

- Birth cohorts

- Research Cohort by Data Linkage

### Domain of ND

Information on the domain of ND, categorised based on the type of data collected, the measure/scale used (e.g. psychometric questionnaires, diagnostic codes) and the approach to measurement (e.g. parent completed questionnaire, clinician judgement) were recorded for each data source. This information was extracted by authors JG and RB from information publicly available relating to the data source such as a website or publications.

   Ethical approval was not required for this study.

## Results

Fifty data sources with ND outcomes were listed in the EUROmediSAFE inventory and contacts in EUROCAT/EUROmediCAT and Euro-Peristat identified an additional 27 data sources. The literature search resulted in 2,798 citations. Based on manual review of the abstract, articles were excluded when related to pre-clinical, genetic, epigenetic, case reports, case series, where the outcome was out of scope (e.g. child's depression; imaging, post-partum depression, pulmonary hypertension), or the exposures / intervention were not SSRIs (other drugs, stress, smoking). Thirteen data sources were identified in the literature (8 of these had already been identified in the EUROmediSAFE inventory or by contacts). An additional 8 data sources were identified by web searches.

   In total 90 data sources were identified across 14 countries. The majority (63.3%) were created for health surveillance and research with the remaining serving administrative purposes (21.1% healthcare databases and 15.6% other administrative databases). As can be seen in Fig 1, half of the data sources were birth cohorts. While the other types of data source were less numerous, these generally included much larger populations than the birth cohorts, for some the entire population of a country, and so represented a much larger sample.

   Across the data sources identified the ND outcomes available were categorised, based on the type of data collected, into five domains of ND—infant development, child behaviour, cognition, educational achievement, and the presence of diagnostic codes for neurodevelopmental disorders. There is inevitable overlap between certain categories but this classification system allows users to select data sources by area of ND which may be relevant to their investigations. In 39 data sources, across 12 countries, it is possible to examine more than one domain of ND.

   Information on infant development was available in 36 data sources across 13 countries, see Table 1, and was recorded in all types of databases except for the education and health insurance claims databases. Assessment of infant development varied, based on clinician

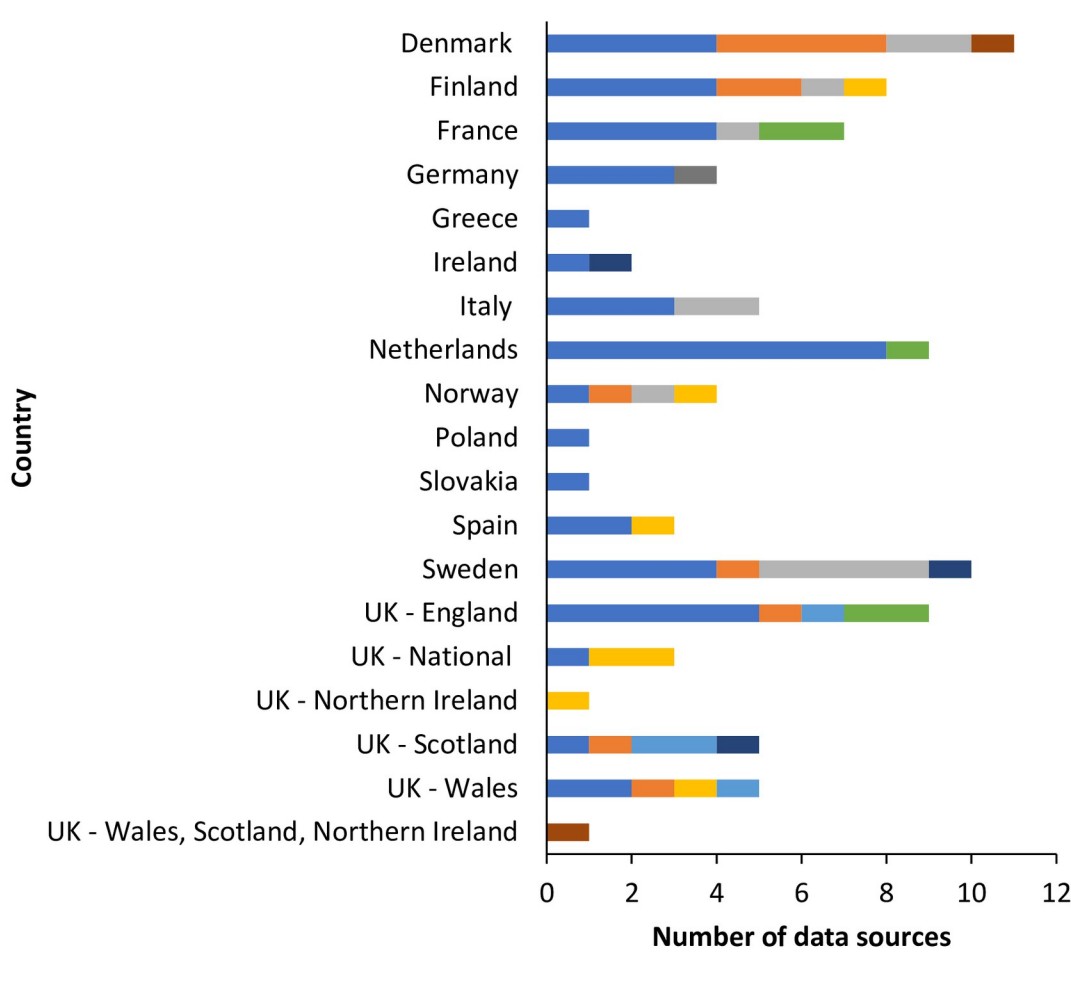

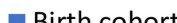

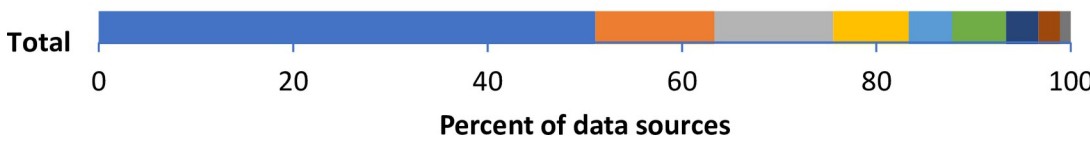

**Fig 1. Type of data source identified across each country and percent of all data sources by type.**

judgement/routine health care, direct/objective assessment and/or parental completed questionnaires. In the birth cohorts' bespoke questionnaires and a wide range of recognised measures, such as the Ages and Stages Questionnaire, Bayley Scales of Infant Development and Denver Developmental Screening Test were used to assess infant development. Such measures were also available in some of the non-cohort data sources such as the child surveillance and disease registries. Assessments made as part of routine healthcare were the predominant source of information in the other types of data source. Here infant development was assessed in routine health and developmental evaluations, service use records, READ (a coding system used in UK primary care) and ICD-10 (International Classification of Disease 10th edition) diagnosis codes, health visitor records and records of referral for support relating to developmental delay.

Assessments of child behaviour were available in 27 data sources across 10 countries, recorded in birth cohorts and child surveillance databases, see Table 2. Behaviour was based on child self-report, direct/objective assessment, parent completed questionnaire/report and teacher review/routine education. Behaviour was assessed using a variety of measurements/scales such as the Strengths & Difficulties Questionnaire, Child Behavior Checklist, Child Behavior Questionnaire, and bespoke questionnaires.

Measurements of cognition (e.g. intelligence, attention, language, memory skills) were available in 29 data sources across 12 countries, recorded in birth cohorts, a child surveillance database, and a register of disability, see Table 3. Cognition was assessed through a varied set of approaches including clinician judgement/routine health care, direct/objective assessment, parent completed questionnaire/report and teacher review/ routine education. As well as a wide range of recognised measures such as the British Ability Scales and Weschler Intelligence Scale for Children, cognitive difficulties could also be identified through records of referral for services relating to cognitive difficulties.

Educational related outcomes were available in 20 data sources across 7 countries, recorded in educational databases, birth cohorts and child surveillance databases, see Table 4. All assessments were based on teacher review or routine educational data or requirement for specialist educational support. In addition to routine educational outcomes, a cohort study and a child surveillance database also had teacher Special Educational Needs ratings available.

Neurodevelopmental disorder diagnostic codes were available in 42 data sources across 11 countries, see Table 5. The presence of ND disorders was based on parent completed questionnaire/report, direct/objective assessment, child self-report, clinician judgement/routine health care, teacher review/routine education and direct/objective assessment. Diagnoses recorded using ICD-9, ICD-10, ICPC (International Classification for Primary Care), DSM (Diagnostic and Statistical Manual of Mental Disorders) IV, and Read codes were available in all database types except for education. ND measures such as the Checklist for autism in toddlers (CHAT), Childhood Asperger Syndrome Test (CAST) and Attention Deficit/Hyperactivity Disorder Test (ADHD) to identify children with autism, Asperger syndrome or ADHD were exclusively found in birth cohorts, although one child surveillance database had results for the Modified Checklist for Autism in Toddlers (M-CHAT).

## Discussion

Pharmacoepidemiologic investigations into ND outcomes in children exposed to a medication during pregnancy lag behind initiatives to understand risk of congenital anomaly. We

**Table 1. Data sources which record infant development.**

| Country | Geographic coverage | Database name | Sub-type of data | Approach to Measurement | Measure/Scale Used^ |
|---------|--------------------|--------------|----------------|----------------------|-------------------|
| Denmark | Births at Skejby Hospital, Denmark | Aarhus Birth Cohort | Birth cohort | Parent complete questionnaire/ report | ASQ |
| Denmark | Copenhagen County | Copenhagen Child Cohort 2000 (CCC2000) | Birth cohort | Direct/Objective Assessment | BSID-II (1.5 years) |
| | | | | Clinician judgement/ routine health care | Health visitor records |
| Denmark | National | Danish National Birth Cohort | Birth cohort | Parent complete questionnaire/ report | Bespoke Questionnaire; Developmental Coordination Disorder Questionnaire |
| Finland | Northern Finland | Northern Finland birth cohort of 1966 | Birth cohort | Parent complete questionnaire/ report | Bespoke Questionnaire |
| France | Haute-Garonne (south-west France) | EFEMERIS (Evaluation in Pregnant Women of MEdicaments and their RISK) | Research Cohort by Data Linkage | Clinician judgement/ routine health care | Bespoke Questionnaire |
| France | National | EPIPAGE 2 Cohort Study | Birth cohort | Parent complete questionnaire/ report | ASQ |
| | | | | Clinician judgement/ routine health care | GMFCS; SCPE diagnostic criteria |
| France | National | Etude Longitudinale Francaise depuis l'Enfance (ELFE) | Birth cohort | Parent complete questionnaire/ report | CDI |
| France | Haute-Garonne (south-west France) | POMME (PrescriptiOn Médicaments Mères Enfants) | Research Cohort by Data Linkage | Clinician judgement/ routine health care | Routine examination |
| Germany | Leipzig | LIFE Child | Birth cohort | Direct/Objective Assessment | BSID-III |
| Greece | Crete | Mother Child Cohort in Crete (RHEA) | Birth cohort | Direct/Objective Assessment | BSID-III |
| Ireland | County Cork | BASELINE: Babies after SCOPE | Birth cohort | Direct/Objective Assessment | BSID |
| Italy | National | Nascita e INFanzia: gli Effetti dell'Ambiente (NINFEA) | Birth cohort | Parent complete questionnaire/ report | Bespoke questionnaire |
| Italy | Florence, Rome, Trieste, Turin and Viareggio | Piccolipiù | Birth cohort | Unclear | Unclear |
| Italy | Emilia Romagna Region | SINPIA ER-Flusso informativo per i servizi di neuropsichiatria infantile dell'infanzia e dell'adolescenza dell'Emilia Romagna | Hospital database* | Clinician judgement/ routine health care | ICD-10; service use records |
| Netherlands | Rotterdam | Generation R | Birth cohort | Direct/Objective Assessment | CDI; MB-CDI Short Form |
| Netherlands | Rotterdam | Generation R Next | Birth cohort | Direct/Objective Assessment | Eye-tracking |
| Netherlands | Westelijke Mijnstreek region | LucKi Birth Cohort Study | Birth cohort | Direct/Objective Assessment | Van Wiechen classification of psychomotor development; Unknown language assessment |
| Netherlands | National | PRIDE Study (PRIDE: PRegnancy and Infant DEvelopment) | Birth cohort | Parent complete questionnaire/ report | ASQ |

*(Continued)*

**Table 1.** (*Continued*)

| Country | Geographic coverage | Database name | Sub-type of data | Approach to Measurement | Measure/Scale Used^ |
|---|---|---|---|---|---|
| **Netherlands** | five municipalities in the North of The Netherlands | Tracking Adolescents' Individual Lives Survey (TRAILS) NEXT | Birth cohort | Direct/Objective Assessment | Bespoke observation and tasks |
| | | | | Parent complete questionnaire/ report | Bespoke questionnaire |
| **Norway** | National | Norwegian Mother, Father and Child Cohort Study (MoBa) | Birth cohort | Parent complete questionnaire/ report | ASQ; Dale sentence complexity task; NVCC; SCQ |
| **Poland** | Eight regions of Poland | REPRO_PL Polish Mother and Child Cohort Study | Birth cohort | Direct/Objective Assessment | BSID-III |
| **Slovakia** | Eastern Slovakia: Michalovce | Slovak PCB study Exposure to polychlorinated biphenyl | Birth cohort | Direct/Objective Assessment | BSID-II |
| **Spain** | Seven Spanish regions (Ribera d'Ebre, Menorca, Granada, Valencia, Sabadell, Asturias, and Gipuzkoa) | INMA-Environment and Childhood Project (INMA Project) | Birth cohort | Direct/Objective Assessment | BSID; Dubowitz Developmental Screening Test |
| **UK—England** | Avon, England | ALSPAC-G2 (second generation of The Avon Longitudinal Study of Parents and Children) | Birth cohort | Parent complete questionnaire/ report | Bespoke developmental questionnaire |
| **UK—England** | Avon, England | Avon Longitudinal Study of Parents & Children/Children of the 90s (ALSPAC) | Birth cohort | Parent complete questionnaire/ report | Denver Developmental Screening Test |
| **UK—England** | Bradford, England | Born in Bradford/Born in Bradford Growing up | Birth cohort | Direct/Objective Assessment | CKAT |
| **UK—England** | England | Community Services Data Set (CSDS) | Child surveillance databases | Parent complete questionnaire/ report | ASQ |
| **UK—England** | Southampton | Southampton Women's Survey | Birth cohort | Direct/Objective Assessment | WPPSI; CANTAB |
| **UK—England** | Wirral, England | Wirral Child Health and Development Study | Birth cohort | Direct/Objective Assessment | BSID-III; NBAS; LabTAB; Physiological responses |
| **UK—Northern Ireland** | Northern Ireland | General Practitioner Information Platform | Primary care database | Clinician judgement/ routine health care | Read codes |
| **UK—Scotland** | Scotland | Child Health Systems Programme—Pre-School (CHSP Pre-School) | Child surveillance databases | Parent complete questionnaire/ report | ASQ; PEDS; PEDS:DM; SOGS II; SSLM. For subset: M-CHAT, PEDS, PEDS:DM, SOGS II, SSLM, Eyberg Child Behaviour Inventory, |
| **UK—Scotland** | Scotland | Growing up in Scotland (GUS) | Birth cohort | Parent complete questionnaire/ report | Bespoke milestone questionnaire |
| **UK—Scotland** | Scotland | Support Needs System (SNS) | Register of disability | Clinician judgement/ routine health care | Referrals |
| **UK—Wales** | Swansea, Wales | Growing up in Wales | Birth cohort | Parent complete questionnaire/ report | Bespoke questionnaire |
| **UK—Wales** | Wales | National Community Child Health Database | Child surveillance databases | Clinician judgement/ routine health care | Routine health and developmental evaluations |

(*Continued*)

**Table 1.** (Continued)

| Country | Geographic coverage | Database name | Sub-type of data | Approach to Measurement | Measure/Scale Used^ |
|---|---|---|---|---|---|
| **UK—Wales, Scotland, Northern Ireland** | Wales, Scotland, Northern Ireland | The National Neonatal Research database (NNRD) | Disease registry | Parent complete questionnaire/ report | Bespoke questionnaire |
| | | | | Direct/Objective Assessment | BSID-III; Griffiths Test; SGS |

* Admission, Episode, Discharge

^See S3 File for abbreviations of ND measurement tools

identified 90 data sources, across 14 countries, which contain information on five domains of ND–infant development, child behaviour, cognitive, education and ND disorders. It is hoped that this inventory of potentially linkable data sources will expedite investigations into risk of ND outcomes in children associated with medication exposures in pregnancy. When selecting data sources for such research it is important to consider the ND outcome reported, the sensitivity and specificity of measurement, variability in measurement approach and the trajectory of skill development. For certain ND outcomes there is continued progress into the second decade of life [27], due to continued development of the architecture in regions of the brain [28]. Each of these is discussed in more detail below.

Although we grouped the ND outcomes available in the data sources into five domains, the measure/scale used, including the presence or absence of medical diagnoses, results of psychometric instruments (questionnaires or tests) completed by parents, teachers, or health care professionals, educational assessments, and registration of children in disability registers and how these were recorded, these varied within each domain. With such variation, the groups are intended to be informative, highlighting data which could be available for that ND outcome. It does not mean that data are collected in a similar enough manner across data sources to be combined directly in analysis. For example, within the cognitive domain there were continually measured IQ scores as well as ICD-10 and other diagnostic codes relating to intellectual disability. The former was measured with a variety of different measures on a continuous scale, yet the latter represents a diagnosis which is likely to only refer to the most severe cases of intellectual difficulties [29, 30]. The suggested groups should be used in future initiatives to direct researchers to data sets that may be available, but mapping exercises, with expert input, will be required to understand the comparability of the data available within each of the data sources for specific research questions.

ND outcomes recorded in birth cohorts and child surveillance databases were more likely to be based on psychometric tests performed on all children included in the cohort. ND outcomes recorded in primary care databases, hospital databases and administrative health insurance claims databases were nearly always based on diagnostic codes. Frequently, ND outcomes based on psychometric instruments were recorded on a continuous scale, increasing the sensitivity and specificity of the outcome data collected, as the functioning of the entire cohort is available. Higher measurement sensitivity reduces the required cohort size and therefore smaller sized cohorts with these measurements can be useful sources. When using diagnostic codes as a marker of the presence or absence of a diagnosis (i.e., ASD or ADHD) it must be recognised that these are based on routine care practices, which only capture the most affected individuals and are only truly accurate in those who were formally reviewed for the diagnosis. The level of social communication and interaction ability in those without the diagnostic code for autism for example is unknown and there is clear evidence that diagnostic processes are

**Table 2. Data sources which record child behaviour.**

| Country | Geographic coverage | Database name | Sub-type of data | Approach to Measurement | Measure/Scale Used^ |
|---|---|---|---|---|---|
| **Denmark** | Copenhagen County | Copenhagen Child Cohort 2000 (CCC2000) | Birth cohort | Parent complete questionnaire/report | CBCL/1.5–5, SDQ, ITSCL |
| **Denmark** | National | Danish National Birth Cohort | Birth cohort | Child Self-Report | SDQ |
| | | | | Parent complete questionnaire/report | SDQ |
| | | | | Teacher Review/ Routine Education | SDQ |
| **Denmark** | Municipality of Odense | Odense Child Cohort | Birth cohort | Parent complete questionnaire/report | CBCL/1.5–5; CBCL/6-18, SRS |
| **Finland** | Southwest Finland Hospital District and the Åland Islands | FinnBrain Birth Cohort Study (FinnBrain) | Birth cohort | Parent complete questionnaire/report | IBQ-R; ECBQ-R |
| | | | | Direct/Objective Assessment | Lab-TAB |
| **Finland** | Helsinki | Perinatal Adverse events and Special Trends in Cognitive Trajectory (PLASTICITY) | Birth cohort | Parent complete questionnaire/report | CBCL |
| | | | | Child Self-Report | CBCL-YSR, BS |
| **Finland** | | Prediction and Prevention of Preeclampsia and Intrauterine Growth Restriction (PREDO) | Birth cohort | Parent complete questionnaire/report | CBCL/1.5–5 |
| **France** | Nancy and Poitiers | EDEN—Study on the pre and early postnatal determinants of child health and development | Birth cohort | Parent complete questionnaire/report | SDQ; EAS |
| **France** | Brittany | PELAGIE study (Endocrine Disruptors: Longitudinal Study on Anomalies in Pregnancy, Infertility and Childhood) | Birth cohort | Parent complete questionnaire/report | SDQ |
| **Germany** | Munich, Leipzig, Wesel, and Bad Honnef, Germany | Influence of life-style factors on the development of the immune system and allergies in East and West Germany (LISA PLUS) | Birth cohort | Parent complete questionnaire/report | SDQ |
| **Germany** | Leipzig | LIFE Child | Birth cohort | Parent complete questionnaire/report | SDQ (10–18); Bespoke Hyperkinetic Questionnaire |
| **Ireland** | County Cork | BASELINE: Babies after SCOPE | Birth cohort | Parent complete questionnaire/report | CBCL, Greenspan Social-Emotional Growth Chart |
| **Italy** | National | Nascita e INFanzia: gli Effetti dell'Ambiente (NINFEA) | Birth cohort | Parent complete questionnaire/report | Bespoke questionnaire; SDQ |
| **Netherlands** | Amsterdam | Amsterdam Born Children and their Development (ABCD) | Birth cohort | Teacher Review/ Routine Education | SDQ |
| | | | | Parent complete questionnaire/report | SDQ; Bespoke Questionnaire |
| **Netherlands** | Drenthe | GECKO Drenthe cohort (GECKO Drenthe) | Birth cohort | Parent complete questionnaire/report | SDQ (Dutch) |
| **Netherlands** | Rotterdam | Generation R | Birth cohort | Parent complete questionnaire/report | Bespoke (inc. IBQ-R, CBQ); ICU |
| **Netherlands** | Westelijke Mijnstreek region | LucKi Birth Cohort Study | Birth cohort | Parent complete questionnaire/report | Unclear |
| **Netherlands** | Five municipalities in the North of the Netherlands | Tracking Adolescents' Individual Lives Survey (TRAILS) NEXT | Birth cohort | Parent complete questionnaire/report | Bespoke questionnaire |
| **Norway** | National | Norwegian Mother, Father and Child Cohort Study (MoBa) | Birth cohort | Parent complete questionnaire/report | CBCL; ICQ-6; EAS; SDQ; ITSEA; PPBS, RS-DBD |
| **Sweden** | Sweden | Child and Adolescent Twin Study in Sweden-CATSS | Birth cohort | Parent complete questionnaire/report | TCI/TCI(J); Youth Psychopathy Inventory; Child Monitoring Scale |

(*Continued*)

**Table 2.** (Continued)

| Country | Geographic coverage | Database name | Sub-type of data | Approach to Measurement | Measure/Scale Used^ |
|---------|---------------------|---------------|------------------|-------------------------|---------------------|
| UK—England | Bradford, England | Born in Bradford/Born in Bradford Growing up | Birth cohort | Parent complete questionnaire/report | SDQ |
| UK—England | Southampton | Southampton Women's Survey | Birth cohort | Parent complete questionnaire/report | SDQ |
| UK–England | Wirral, England | Wirral Child Health and Development Study | Birth cohort | Teacher Review/Routine Education | CBCL-TRF; SDQ |
| | | | | Parent complete questionnaire/report | IBQ-R; ECBQ; CBQ; BITSEA; CBCL; SDQ |
| UK—National | National | Millennium Cohort Study | Birth cohort | Parent complete questionnaire/report | SDQ |
| UK—Scotland | Scotland | Child Health Systems Programme—Pre-School (CHSP Pre-School) | Child surveillance databases | Parent complete questionnaire/report | Eyberg Child Behaviour Inventory |
| UK—Scotland | Scotland | Growing up in Scotland (GUS) | Birth cohort | Parent complete questionnaire/report | SDQ; CBQ; Pre-School Activities Inventory |
| UK—Wales | Cardiff, Wales | Cardiff Child Development Study | Birth cohort | Parent complete questionnaire/report | IBQ |
| UK—Wales | Wales | National Community Child Health Database | Child surveillance databases | Teacher Review/Routine Education | Teacher behaviour ratings |

^See S3 File for abbreviations of ND measurement tools

influenced by family background, ethnicity, parental education and socioeconomic status [31, 32]. Thus, children may experience moderate levels of disruption of function but may not either reach diagnostic thresholds or never be reviewed for the diagnostic code in question. Therefore, different data sources may be utilised in different ways, to answer different questions and both will have their inherent strengths and limitations about the sensitivity and specificity of the measurement of the ND outcome.

Both the approaches to measuring ND outcomes include variability in assessment of ND outcomes. The birth cohorts and child surveillance datasets included a broad selection of standardised psychometric instruments e.g., Bayley Scales of Infant Development to assess early development, or Wechsler Preschool and Primary Scale of Intelligence to assess child IQ. National diversity in healthcare provision and practice also contributes to variability when combining diagnostic data across healthcare systems and countries due to variability in regional and national approaches to diagnosis [33]. It should also be recognised that the diagnostic criteria for certain disorders has varied over time and across countries. Variability needs to be considered by the users of the data sources and steps taken to maximise the comparability of data. For some data sources such as the primary care databases, which do not tend to be standardised, this may include standardisation and validation of the data before they can be used. This increases the time and cost of using the data and requires collaboration with local data providers and experts [34].

The heterogeneity is even more pronounced when it comes to educational outcomes, where educational systems and teacher's assessment are country specific. One way to report a common indicator is to assess the proportion of children above or below the average or the proportion of children considered to pass a specific routine exam. However, this proposal is not without difficulties, as children sit formal examinations at different ages across Europe. For instance, in the UK, children are tested at 4 key stages (ages 5–7, 8–11, 12–14, and 15–16). In

**Table 3. Data sources which record cognitive outcomes.**

| Country | Geographic coverage | Database name | Sub-type of data | Approach to Measurement | Measure/Scale Used^ |
|---|---|---|---|---|---|
| **Denmark** | Copenhagen County | Copenhagen Child Cohort 2000 (CCC2000) | Birth cohort | Direct/Objective Assessment | WISC IV (1 subtest) |
| **Denmark** | Municipality of Odense | Odense Child Cohort | Birth cohort | Direct/Objective Assessment | WISC-V (4 subtest version) |
| **Finland** | Southwest Finland Hospital District and the Åland Islands | FinnBrain Birth Cohort Study (FinnBrain) | Birth cohort | Unclear | Unclear |
| **Finland** | Northern Finland | Northern Finland birth cohort of 1966 | Birth cohort | Direct/Objective Assessment | WISC |
| **Finland** | Helsinki | Perinatal Adverse events and Special Trends in Cognitive Trajectory (PLASTICITY) | Birth cohort | Direct/Objective Assessment | ITPA, WISC, WAIS, WMS, Test of Motor Impairment, Michelsson Neurodevelopmental Screen, Benton Visual Memory Test, Goodenough Drawing Test, Frostig Test of Visual Perception, Dubowitz Developmental Screening Test, Tapping |
| **Finland** | 10 study hospitals (Jorvi Hospital in Espoo, the Women's Hospital and the Kätilöopisto Maternity Hospital in Helsinki, the Hyvinkää Hospital in Hyvinkää, the Kanta-Häme Central Hospital in Hämeenlinna, the Iisalmi Hospital in Iisalmi, the North Karelia Central Hospital in Joensuu, the Kuopio University Hospital in Kuopio, the Päijät-Häme Central Hospital in Lahti and the Tampere University Hospital in Tampere) | Prediction and Prevention of Preeclampsia and Intrauterine Growth Restriction (PREDO) | Birth cohort | Parent complete questionnaire/ report | ASQ-III |
| **France** | National | Etude Longitudinale Francaise depuis l'Enfance (ELFE) | Birth cohort | Direct/Objective Assessment | BAS-II; BSRA; WISC |
| | | | | Parent complete questionnaire/ report | MB-CDI |
| **France** | Brittany | PELAGIE study (Endocrine Disruptors: Longitudinal Study on Anomalies in Pregnancy, Infertility and Childhood) | Birth cohort | Direct/Objective Assessment | WISC (subtests); Visual go/no go task |
| **Germany** | Munich and Nuremberg | Childhood Obesity—Early Programming by Infant Nutrition (CHOPIN) | Birth cohort | Unclear | Unclear |
| **Greece** | Crete | Mother Child Cohort in Crete (RHEA) | Birth cohort | Parent complete questionnaire/ report | CBCL/6-18; SDQ; ADHDT |
| | | | | Direct/Objective Assessment | MSCA; N-Back; Attention Network Test; Trail Making Test; Raven's Test |
| **Ireland** | County Cork | BASELINE: Babies after SCOPE | Birth cohort | Direct/Objective Assessment | Kaufman Intelligence Test-II |
| **Italy** | eight Italian hospitals | Multiple Births Cohort Study (MUBICOS) | Birth cohort | Unclear | Unclear |
| **Netherlands** | Amsterdam | Amsterdam Born Children and their Development (ABCD) | Birth cohort | Direct/Objective Assessment | Amsterdam Neuropsychological Tasks |
| **Netherlands** | Rotterdam | Generation R | Birth cohort | Direct/Objective Assessment | NEPSY-II; BRIEF; Snijders-Oomen Non-Vernal Intelligence Test |
| **Netherlands** | Utrecht and its surrounding areas | YOUth Cohort study | Birth cohort | Direct/Objective Assessment | Penn word memory, Penn motor praxis test, WISC-V |

(*Continued*)

**Table 3.** (Continued)

| Country | Geographic coverage | Database name | Sub-type of data | Approach to Measurement | Measure/Scale Used^ |
|---|---|---|---|---|---|
| **Norway** | National | Norwegian Mother, Father and Child Cohort Study (MoBa) | Birth cohort | Parent complete questionnaire/ report | CDI; SLAS; CCC-2; Sprak20; EDI |
| **Spain** | Seven Spanish regions (Ribera d'Ebre, Menorca, Granada, Valencia, Sabadell, Asturias, and Gipuzkoa) | INMA-Environment and Childhood Project (INMA Project) | Birth cohort | Direct/Objective Assessment | MSCA; K-CPT; Batelle Developmental Inventory; California Preschool Social Competence Scale; |
| **Spain** | Health Area I, VI and VII of the Region of Murcia | NELA—Nutrition in Early Life and Asthma (NELA) | Birth cohort | Unclear | Unclear |
| **Sweden** | Sweden | Child and Adolescent Twin Study in Sweden- CATSS | Birth cohort | Direct/Objective Assessment | WISC-IV; CGAS |
| **Sweden** | Stockholm County | Habilitation Register | Register of disability | Clinician judgement/ routine health care | Service referrals |
| **Sweden** | Stockholm County | Stockholm Youth Cohort | Research Cohort by Data Linkage | Clinician judgement/ routine health care | ICD-10 codes |
| **Sweden** | Värmland county | Swedish Environmental Longitudinal, Mother and child, Asthma and allergy study | Birth cohort | Direct/Objective Assessment | Swedish Language Development Scale |
| **UK—England** | Avon, England | Avon Longitudinal Study of Parents & Children/Children of the 90s (ALSPAC) | Birth cohort | Direct/Objective Assessment | WPPSI; WISC; Griffiths Test |
| **UK—England** | Bradford, England | Born in Bradford/Born in Bradford Growing up | Birth cohort | Direct/Objective Assessment | BPVS |
| **UK—England** | Wirral, England | Wirral Child Health and Development Study | Birth cohort | Direct/Objective Assessment | CANTAB (IED, SWM, SOC); BPVS; WASI; BAS; Executive Function battery; Socio-Emotional Battery |
| **UK—National** | National | Millennium Cohort Study | Birth cohort | Direct/Objective Assessment | BAS-II, All Wales Reading Test, CANTAB (SWM/SOC) |
| **UK—Scotland** | Scotland | Growing up in Scotland (GUS) | Birth cohort | Direct/Objective Assessment | BAS; Children's Communication Checklist |
| **UK—Wales** | Cardiff, Wales | Cardiff Child Development Study | Birth cohort | Direct/Objective Assessment | Self-regulation battery: Tower of Cardiff; Snack Delay; Whisper Task; Nonverbal Stroop card sorting test, Amsterdam Neuropsychological Tasks. |
| **UK—Wales** | Wales | National Community Child Health Database | Child surveillance databases | Teacher Review/ Routine Education | Teacher developmental ratings |

^See S3 File for abbreviations of ND measurement tools

contrast, children in Finland sit their first examinations at age 16 years. Educational data and child health surveillance data have been used much less frequently to determine ND risk than other types of data. However, they have the benefit that data are available for the whole population, not just those referred with a suspected diagnosis, and represent some domains of ND over the long-term/teenage years.

Finally, it should be considered that all data sources have potential for bias. In countries or regions with health registries, it may be comparatively cheap and fast to use diagnosis codes. However, detection bias cannot be ruled out. For example, it is not possible to blind a child's exposure status from the health care professionals reviewing the child to rate the ND outcome.

**Table 4. Data sources which record educational outcomes.**

| Country | Geographic coverage | Database name | Type of data | Approach to Measurement | Measure/Scale Used^ |
|---|---|---|---|---|---|
| **Denmark** | National | Academic Achievement Register (AAR) | Educational database | Teacher Review/ Routine Education | Routine Educational Outcomes |
| **Denmark** | National | Population's Education Register (PER) | Educational database | Teacher Review/ Routine Education | Routine Educational Outcomes |
| **Denmark** | National | Student Register 1 | Educational database | Teacher Review/ Routine Education | Routine Educational Outcomes |
| **Denmark** | National | Student Register 2 | Educational database | Teacher Review/ Routine Education | Routine Educational Outcomes |
| **Finland** | National | Discontinuation of education | Educational database | Teacher Review/ Routine Education | Routine Educational Outcomes |
| **Finland** | Helsinki | Perinatal Adverse events and Special Trends in Cognitive Trajectory (PLASTICITY) | Birth cohort | Teacher Review/ Routine Education | Routine Educational Outcomes |
| **Finland** | National | Upper secondary general school education | Educational database | Teacher Review/ Routine Education | Routine Educational Outcomes |
| **Italy** | National | Nascita e INFanzia: gli Effetti dell'Ambiente (NINFEA) | Birth cohort | Teacher Review/ Routine Education | Parent reported grades |
| **Netherlands** | Amsterdam | Amsterdam Born Children and their Development (ABCD) | Birth cohort | Teacher Review/ Routine Education | CITO Index |
| **Norway** | National | National Education Database NUDB. | Educational database | Teacher Review/ Routine Education | Routine educational outcomes |
| **Sweden** | National | The Swedish Register of Education | Educational database | Teacher Review/ Routine Education | Routine education outcomes |
| **UK—England** | Avon, England | Avon Longitudinal Study of Parents & Children/Children of the 90s (ALSPAC) | Birth cohort | Teacher Review/ Routine Education | Routine education outcomes; teacher rated questionnaires |
| **UK—England** | Bradford, England | Born in Bradford/Born in Bradford Growing up | Birth cohort | Teacher Review/ Routine Education | Routine education outcomes; local authority data |
| **UK—England** | England | Community Services Data Set (CSDS) | Child surveillance databases | Teacher Review/ Routine Education | Teacher Special Educational Need (SEN) ratings |
| **UK—England** | Two acute and one Mental Health Care National Health Service (NHS) Provider in South London | Early Life Cross Linkage in Research (eLIXIR) Partnership | Research Cohort by Data Linkage | Teacher Review/ Routine Education | Routine education outcomes |
| **UK—England** | England | National Pupil Database | Educational database | Teacher Review/ Routine Education | Routine education outcomes |
| **UK—National** | National | Millennium Cohort Study | Birth cohort | Teacher Review/ Routine Education | Teacher Special Educational Need (SEN) ratings; Routine education outcomes |
| **UK—Scotland** | Scotland | Achievement of Curriculum for Excellence Levels | Educational database | Teacher Review/ Routine Education | Routine education outcomes |
| **UK—Wales** | Wales | Education Attainment | Educational database | Teacher Review/ Routine Education | Routine education outcomes |
| **UK—Wales** | Wales | National Community Child Health Database | Child surveillance databases | Teacher Review/ Routine Education | Teacher SEN ratings |

^See S3 File for abbreviations of ND measurement tools

Whilst, this may be less of an influence early on, once an association between a medication exposure and a child ND outcome has been established this may positively bias practice. For example, only prescribing this medication to clinically severe patients whose disease cannot be controlled with less teratogenic or toxic medications. Additional biases come from population health behaviour. For instance, it is suspected that women exposed to a suspected teratogen, or

**Table 5. Data sources which record neurodevelopmental disorders.**

| Country | Geographic coverage | Database name | Sub-type of data | Approach to Measurement | Measure/Scale Used^ |
|---------|--------------------|--------------|-----------------|------------------------|--------------------|
| **Denmark** | National | ADHD Database | Disease registry | Clinician judgement/ routine health care | ICD codes |
| **Denmark** | National (Denmark, Greenland and the Faroes) | Central Psychiatric Register | Hospital databases* | Clinician judgement/ routine health care | ICD Codes |
| **Denmark** | Copenhagen County | Copenhagen Child Cohort 2000 (CCC2000) | Cohort study | Parent complete questionnaire/ report | CHAT, DAWBA |
| | | | | Clinician judgement/ routine health care | ICD-10, DSM IV codes |
| **Denmark** | National | National Patient Register | Hospital databases* | Clinician judgement/ routine health care | ICD codes |
| **Denmark** | Municipality of Odense | Odense Child Cohort | Cohort study | Parent complete questionnaire/ report | SRS; ADHD-Rating Scale |
| **Finland** | National | Care Register for Health Care (HILMO) (replaced the Hospital Discharge Register in 1994) | Hospital databases* | Clinician judgement/ routine health care | ICD-9; ICD-10 in recent years |
| **Finland** | Northern Finland | Northern Finland birth cohort of 1966 | Cohort study | Direct/Objective Assessment | WISC |
| **Finland** | National | Primary health care (AvoHILMO) | Primary care database | Clinician judgement/ routine health care | ICD-10/ICPC codes |
| **France** | National | French national health data system (SNDS), health insurance claim and hospital discharge databases | Hospital database | Clinician judgement/ routine health care | ICD-10 codes |
| **Germany** | Munich and Nuremberg | Childhood Obesity—Early Programming by Infant Nutrition (CHOPIN) | Cohort study | Unclear | Unclear |
| **Germany** | 17% of national population | German Pharmacoepidemiological Research Database (GePaRD): Hospital data and Outpatient data | Administrative health insurance claims database | Clinician judgement/ routine health care | ICD-10 codes |
| **Ireland** | National | National Ability Supports System (created in 2018 by merging National Intellectual Disability Database (NIID) and National Physical and Sensory Disability Database (NPSDD)) | Register of disability | Clinician judgement/ routine health care | ICD-10; service use records |
| **Italy** | Florence, Rome, Trieste, Turin and Viareggio | Piccolipiù | Cohort study | Unclear | Unclear |
| **Italy** | Emilia Romagna Region | SINPIA ER-Flusso informativo per i servizi di neuropsichiatria infantile dell'infanzia e dell'adolescenza dell'Emilia Romagna | Hospital databases* | Clinician judgement/ routine health care | ICD-10; service use records |

(*Continued*)

**Table 5.** (Continued)

| Country | Geographic coverage | Database name | Sub-type of data | Approach to Measurement | Measure/Scale Used^ |
|---|---|---|---|---|---|
| **Italy** | Tuscany | Tuscany SALM–mental health services | Hospital databases* | Clinician judgement/ routine health care | ICD-10 |
| **Netherlands** | Rotterdam | Generation R | Cohort study | Child Self-Report | AQ-Short |
| | | | | Direct/Objective Assessment | Autism Diagnostic Interview—Revised |
| | | | | Parent complete questionnaire/ report | SRS |
| **Netherlands** | 25% of the Netherlands | PHARMO-PRN cohorts | Research Cohort by Data Linkage | Clinician judgement/ routine health care | ICD codes |
| **Netherlands** | National | PRIDE Study (PRIDE: PRegnancy and Infant DEvelopment) | Cohort study | Clinician judgement/ routine health care | Routine medical records |
| **Norway** | National | Norwegian Mother, Father and Child Cohort Study (MoBa) | Cohort study | Parent complete questionnaire/ report | ESAT; M-CHAT; CAST; CPRS-R |
| **Norway** | National | Norwegian Patient Registry (NPR) | Hospital databases* | Clinician judgement/ routine health care | ICD-10 codes |
| **Norway** | National | Norwegian Registry for Primary Health Care | Primary care database | Clinician judgement/ routine health care | ICD-10/ICPC codes |
| **Spain** | Catalonia | Information system for research in primary care (SIDIAP) | Primary care database | Clinician judgement/ routine health care | ICD-10 |
| **Spain** | Seven Spanish regions (Ribera d'Ebre, Menorca, Granada, Valencia, Sabadell, Asturias, and Gipuzkoa) | INMA-Environment and Childhood Project (INMA Project) | Cohort study | Clinician judgement/ routine health care | ADHD Criteria of DSM-IV; CAST |
| **Spain** | Seven Spanish regions (Ribera d'Ebre, Menorca, Granada, Valencia, Sabadell, Asturias, and Gipuzkoa) | INMA-Environment and Childhood Project (INMA Project) | Cohort study | Parent complete questionnaire/ report | ADHD Criteria of DSM-IV; CAST |
| **Sweden** | South East | All Babies in Southeast Sweden (ABIS) | Cohort study | Clinician judgement/ routine health care | ICD-9/ICD-10 codes |
| **Sweden** | Sweden | Child and Adolescent Twin Study in Sweden- CATSS | Cohort study | Child Self-Report | ADHD self-report scale |
| | | | | Clinician judgement/ routine health care | A-TAC; ASDI |
| | | | | Parent complete questionnaire/ report | CBCL |

(*Continued*)

**Table 5.** (Continued)

| Country | Geographic coverage | Database name | Sub-type of data | Approach to Measurement | Measure/Scale Used^ |
|---|---|---|---|---|---|
| **Sweden** | Stockholm County | Clinical Database for Child and Adolescent Psychiatry in Stockholm | Hospital databases* | Clinician judgement/ routine health care | DSM-IV/ICD-10 codes |
| **Sweden** | National | National Patient Register | Hospital databases* | Clinician judgement/ routine health care | ICD-0 codes |
| **Sweden** | Stockholm County | Stockholm Adult Psychiatric Care Register | Hospital databases* | Clinician judgement/ routine health care | DSM-IV/ICD-10 codes |
| **Sweden** | Stockholm County | Stockholm Youth Cohort | Research Cohort by Data Linkage | Clinician judgement/ routine health care | DSM-IV/ICD-10 codes; Service referrals |
| **Sweden** | Stockholm County | VAL database | Hospital databases* | Clinician judgement/ routine health care | ICD-10 codes; referrals |
| **UK—England** | Two acute and one Mental Health Care National Health Service (NHS) Provider in South London | Early Life Cross Linkage in Research (eLIXIR) Partnership | Research Cohort by Data Linkage | Clinician judgement/ routine health care | Read codes |
| **UK—England** | England | ResearchOne database | Research Cohort by Data Linkage | Clinician judgement/ routine health care | ICD-10 codes; referrals (Read codes) |
| **UK—England** | Wirral, England | Wirral Child Health and Development Study | Cohort study | Parent complete questionnaire/ report | DAWBA; Connors Checklist; SCQ |
| **UK—National** | National | Clinical Practice Research Datalink | Primary care database | Clinician judgement/ routine health care | ICD-10 codes, referrals (Read codes |
| **UK—National** | National | The Health Improvement Network (THIN) | Primary care database | Clinician judgement/ routine health care | Read codes; referrals |
| **UK—Northern Ireland** | Northern Ireland | General Practitioner Information Platform | Primary care database | Clinician judgement/ routine health care | Read codes |
| **UK—Scotland** | Scotland | Child Health Systems Programme—Pre-School (CHSP Pre-School) | Child surveillance databases | Parent complete questionnaire/ report | M-CHAT |
| **UK—Scotland** | Scotland | Child Health Systems Programme—School (CHSP School) | Child surveillance databases | Clinician judgement/ routine health care | ICD-10 codes, referrals |
| **UK—Scotland** | Scotland | Support Needs System (SNS) | Register of disability | Clinician judgement/ routine health care | Referrals |

(*Continued*)

**Table 5.** (Continued)

| Country | Geographic coverage | Database name | Sub-type of data | Approach to Measurement | Measure/Scale Used^ |
|---|---|---|---|---|---|
| **UK—Wales** | Cardiff, Wales | Cardiff Child Development Study | Cohort study | Parent complete questionnaire/ report | CBCL/1.5–5; Developmental Milestones Questionnaire; Connors 3 ADHD Index-Parent Report |
| | | | | Teacher Review/ Routine Education | CBCL-TRF |
| | | | | Direct/Objective Assessment | Preschool Age Psychiatric Assessment |
| **UK—Wales** | 70% of Wales | Primary Care GP dataset | Primary care database | Clinician judgement/ routine health care | ICD- 10 codes |
| **UK—Wales, Scotland, Northern Ireland** | Wales, Scotland, Northern Ireland | The National Neonatal Research database (NNRD) | Disease registry | Clinician judgement/ routine health care | Neurological diagnoses |

* Admission, Episode, Discharge

^See S3 File for abbreviations of ND measurement tools

women with a history of mental illness, may be more likely to get health or developmental referrals for their children.

## Strengths and weaknesses

It is not possible to confirm that every data source has been identified and is included in the inventory. This is particularly true for databases that are not included in published papers, databases that are used for medications other than SSRIs or in European countries with no EUROCAT/EUROmediCAT or Euro-Peristat contact available. However, given the global move towards using electronic administrative databases for research, it is likely that we have identified the databases that are most accessible for pharmacoepidemiologic research.

A pre-requisite for inclusion was that a data source should contain information on maternal medication use in pregnancy, either in the data source itself or through linkage with another data source. The level of detail relating to maternal medication use will vary, potentially impacting how useful these data sources are when examining the impact of medication use in pregnancy on ND outcomes. Data sources which can be linked to, for example, large administrative prescribing databases could potentially provide quite detailed information such as specific drug name/code, dates of prescribing/dispensation, dose, and route of administration. However, maternal use of over-the-counter medication would not be available. In contrast, in birth cohorts, where the impact of maternal medication use in pregnancy is not the primary research question, limited maternal medication use may be recorded. For example, exposure to broad drug groups such as antibiotics or anticonvulsants may be recorded rather than the specific drug. However, over the counter medication use may be available.

Data sources were not contacted to confirm if they would allow their data i.e. medication exposure records or ND outcome data to be used in secondary research. In a similar piece of work the response rate was just 52% from data sources contacted [35]. Where data sources, in particular birth cohorts, do not have websites or publicly available documentation relating to

methodology the determination of ND measures available relied on published articles. It may be possible that such data sources hold more data relating to ND outcomes than have been identified. For six birth cohorts it was not possible to determine the methods by which ND was assessed (1 assessing infant development, 1 child behaviour, 4 cognitive and 2 neurodevelopmental disorder assessments).

## Conclusions

Ninety European data sources were identified with potential to be used to assess five domains of ND following maternal medication use during pregnancy. These have great potential to be used in pharmacoepidemiologic research into the safety of SSRIs and other medications in pregnancy potentially associated with ND outcomes in children. Caution must be used when combining varied approaches to ND outcome measurement and consideration regarding the sensitivity and specificity of the outcome measure selected and the age of the child at review/ follow up should be borne in mind. This inventory is an invaluable resource for researchers planning future studies to investigate the ND impact of medication exposures during pregnancy.

## Supporting information

**S1 File. Search query.**
(DOCX)

**S2 File. Types of data source with ND outcomes available.**
(DOCX)

**S3 File. Abbreviations of ND measurement tools.**
(DOCX)

## Acknowledgments

The research leading to these Results was conducted as part of the ConcePTION consortium. This paper only reflects the personal views of the stated authors.

The authors would like to acknowledge their colleagues in EUROCAT, EUROmediCAT and Euro-Peristat for providing information on data sources available in their countries. Professor Helen Dolk for helping to shape the paper and Matthew Bluett Duncan for assistance with the data collation.

## Author Contributions

**Conceptualization:** Joanne Given, Maria Loane.

**Data curation:** Joanne Given, Rebecca L. Bromley, Sandra Lopez-Leon.

**Funding acquisition:** Rebecca L. Bromley, Florence Coste, Maria Loane.

**Investigation:** Joanne Given, Rebecca L. Bromley.

**Methodology:** Joanne Given, Rebecca L. Bromley, Florence Coste, Sandra Lopez-Leon, Maria Loane.

**Supervision:** Maria Loane.

**Validation:** Joanne Given, Rebecca L. Bromley.

**Visualization:** Joanne Given.

**Writing – original draft:** Joanne Given.

**Writing – review & editing:** Joanne Given, Rebecca L. Bromley, Florence Coste, Sandra Lopez-Leon, Maria Loane.

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
