## [Decision Letter · Decision Letter 0]

14 Mar 2022

PONE-D-21-33083An inventory of European data sources to support pharmacoepidemiologic research on neurodevelopmental outcomes in children following medication exposure in pregnancy: A contribution from the ConcePTION projectPLOS ONE

Dear Dr. Loane,

Thank you for submitting your manuscript to PLOS ONE. After careful consideration, we feel that it has merit but does not fully meet PLOS ONE’s publication criteria as it currently stands. Therefore, we invite you to submit a revised version of the manuscript that addresses the points raised during the review process.

We look forward to receiving your revised manuscript.

Kind regards,

Maria Christine Magnus, PhD

Academic Editor

PLOS ONE

Journal Requirements:

2. Please note that in order to use the direct billing option the corresponding author must be affiliated with the chosen institute. Please either amend your manuscript to change the affiliation or corresponding author, or email us at plosone@plos.org with a request to remove this option.

Additional Editor Comments:

We have received comments from two reviewers on your manuscript. I look forwards to receiving your revised manuscript addressing these comments.

Reviewers' comments:

Reviewer's Responses to Questions

**Comments to the Author**

1. Is the manuscript technically sound, and do the data support the conclusions?

Reviewer #1: Yes

Reviewer #2: Yes

2. Has the statistical analysis been performed appropriately and rigorously? 

Reviewer #1: N/A

Reviewer #2: N/A

3. Have the authors made all data underlying the findings in their manuscript fully available?

Reviewer #1: Yes

Reviewer #2: Yes

4. Is the manuscript presented in an intelligible fashion and written in standard English?

Reviewer #1: Yes

Reviewer #2: Yes

5. Review Comments to the Author

Reviewer #1: This review provided a very good overview of the European data sources available for researchers interested in studying neurodevelopmental outcomes related to medication use in pregnancy. I appreciated the way the outcomes were grouped in five broad categories.

Main comments:

1. Material and methods, lines 175-176: It was not clear in the main manuscript what was meant by “prospective studies” that were to be excluded. Birth cohorts are also prospective studies, so it was clearer in the supplementary material (e.g., RCTs). Please explain this exclusion criteria in more detail in the main manuscript.

2. It was clear that diagnosis data would be most relevant for neurodevelopmental disorders (Table 5). However, ICD codes were mentioned once in Table 1 (Italy) and Table 3 (Sweden). Either these should be removed from Tables 1 and 3, or other sources would need to be added, e.g., Norwegian Patient Registry. Please clarify, and ideally, list or give examples of ICD codes relevant for Tables 1 (infant development) and 3 (cognitive outcomes).

3. It would be helpful for the reader to know the source(s) of information for maternal medication. It was reported (lines 400-401) the authors did not check if maternal medication information was available. It seems important to at least know that there is drug data available since this is the reason for compiling this list of data sources. I suggest that the authors add where maternal medication information comes from if it must be linked from another source.

4. Conclusions, line 416: Need to mention specificity. This is perhaps more important than sensitivity when it comes to diagnoses for obtaining an unbiased relative risks.

5. The link to descriptions of the abbreviations in the tables did not work. Therefore, I could not review them. Please make sure these are included in the next version.

6. Table 5: Should also mention ICPC codes for Finland and Norway primary care data

7. In some cases, too many abbreviations are used: e.g. in Table 5, remove LPR for Denmark, NRPHC for Norway, PAR for Sweden. I don’t think these are commonly used and you have the full name written so it seems unnecessary to include.

8. Can you comment on how you had enough information to include the data source in the paper, but not to know how the outcomes were assessed for those where the method to assess ND was unclear?

Minor comments:

9. Table 1: Aarhus, not Aarhous

10. Lines 69, 302: Do not use the acronym CA. It is only mentioned one time after introducing it in the introduction, therefore it is not needed and clearer to write in full.

11. Line 310: add the word “brain” – brain regions is clearer

12. Line 373-4, blinding of the health care professionals: do you mean to the child to the exposure status of the child?

13. Line 414: add the word “potential” – associated with potential ND outcomes or potentially associated

14. Tables 2, 3: Not necessary to list every hospital name of the 10 from the Finnish study called PREDO.

15. Tables: Be consistent when you write ICD-10 or ICD 10 (with or without hyphen)

16. Table 5: Finland HILMO should include ICD-10 in recent years

17. Table 5: Sweden ABIS study – should say Southeast or Southwest in geographic coverage. There is a mismatch. Also, “diagnosis codes” is vague. Is it ICD-10 like other sources?

18. Table 5: Sweden Clinical database for child and adolescent psychiatry in Sweden should say DSM-IV (not DSV-IV)

Reviewer #2: 1. The aim of this paper needs to be justified, given the project has been established.

2. The age of children would be important for the study of neurodevelopmental disorders. Thus, it would be very informative to list the start (end) year of the cohorts and registers to identify the children.

3. L371-379: The authors stated that detection bias is always possible when using health registries. To my knowledge, detection bias is less likely in registers since the prescribers and clinicians who see the children are often different. Moreover, as the author mentioned, a large proportion of women take medications during pregnancy. Therefore, medications do not necessarily influence the doctors’ diagnosis. For instance, clinicians may be more reluctant to prescribe the medication to pregnant women unless they have severe disorders when an association is established. It is difficult to know whether any associations are due to positively biased practice or indication for treatment.

4. There are still cohorts/registers missing from the ConcePTION project. Please clarify whether this project will be static, i.e., restricting to these 90 data sources, or be dynamic by identifying more relevant resources?

6. PLOS authors have the option to publish the peer review history of their article (what does this mean?). If published, this will include your full peer review and any attached files.

Reviewer #1: No

Reviewer #2: No

---

## [Author Response · Author response to Decision Letter 0]

23 Aug 2022

Reviewer #1: This review provided a very good overview of the European data sources available for researchers interested in studying neurodevelopmental outcomes related to medication use in pregnancy. I appreciated the way the outcomes were grouped in five broad categories.

Main comments:

1. Material and methods, lines 175-176: It was not clear in the main manuscript what was meant by “prospective studies” that were to be excluded. Birth cohorts are also prospective studies, so it was clearer in the supplementary material (e.g., RCTs). Please explain this exclusion criteria in more detail in the main manuscript.

We have added more text in the manuscript (line 175) explaining the exclusion: Prospective Studies such as case reports, clinical studies, randomised controlled trials and adverse drug reaction databases were excluded as these have small sample sizes or selected populations.

2. It was clear that diagnosis data would be most relevant for neurodevelopmental disorders (Table 5). However, ICD codes were mentioned once in Table 1 (Italy) and Table 3 (Sweden). Either these should be removed from Tables 1 and 3, or other sources would need to be added, e.g., Norwegian Patient Registry. Please clarify, and ideally, list or give examples of ICD codes relevant for Tables 1 (infant development) and 3 (cognitive outcomes).

We explained in lines 223-228 that there was overlap between the 5 domains of neurodevelopment. Tables 1 to 3 represent a variety of data sources including longitudinal cohorts, and databases which employ a variety of approaches to neurodevelopmental outcome measurement. The majority of sources reported in Tables 1-3 collect data on skill acquisition or functioning (e.g. data collected with the Bayley Scales of Infant and Toddler Development, or the Strengths and Difficulties Questionnaire) rather than the presence or absence of a particular set of symptoms and therefore ICD codes are not used by these sources to summarise their data. We have listed ICD codes, where the source provides them. We think a unique aspect of this manuscript is the provision of data sources which go beyond ICD codes which are a relatively underutilised resource which could be used for pharmacovigilance work in the future. 

.3. It would be helpful for the reader to know the source(s) of information for maternal medication. It was reported (lines 400-401) the authors did not check if maternal medication information was available. It seems important to at least know that there is drug data available since this is the reason for compiling this list of data sources. I suggest that the authors add where maternal medication information comes from if it must be linked from another source.

The reviewer has misinterpreted the meaning of the sentence: “Data sources were not contacted to confirm the availability of medication exposure or ND outcome data for secondary research purposes.”

We know the data are available, but we do not know if the data owners would allow their data to be used for secondary research. There are many reasons why data providers may not allow their data to be used in secondary research:

• Data not collected for research purposes

• Quality of the data

• Potential risks of disclosure, especially if there are small numbers (i.e. rare exposures and rare outcomes)

We have clarified this in the manuscript (line 402): “Data sources were not contacted to confirm if they would allow their data i.e. medication exposure records or ND outcome data to be used in secondary research.”

4. Conclusions, line 416: Need to mention specificity. This is perhaps more important than sensitivity when it comes to diagnoses for obtaining an unbiased relative risks.

We agree with the reviewers and revised the sentence: “ Caution must be used when combining varied approaches to ND outcome measurement and consideration regarding the sensitivity and specificity of the outcome measure selected and the age of the child at review/follow up should be borne in mind.“ We have amended this throughout the paper

5. The link to descriptions of the abbreviations in the tables did not work. Therefore, I could not review them. Please make sure these are included in the next version.

The list of abbreviations were uploaded as Supplementary File 3. We are sorry that the reviewer was unable to access this information, accessible by clicking on the Supplementary File link at the end of the document.

6. Table 5: Should also mention ICPC codes for Finland and Norway primary care data

These have been added. 

7. In some cases, too many abbreviations are used: e.g. in Table 5, remove LPR for Denmark, NRPHC for Norway, PAR for Sweden. I don’t think these are commonly used and you have the full name written so it seems unnecessary to include.

We have deleted these abbreviations as suggested.

 8. Can you comment on how you had enough information to include the data source in the paper, but not to know how the outcomes were assessed for those where the method to assess ND was unclear?

These were five birth cohorts where the articles or websites describing the data list for example ‘questionnaires’ or measurement of ‘cognitive function, language, Autism or ADHD’ but no further detail on what measurement tools were used was available either via the website or in the papers published. 

Minor comments:

9. Table 1: Aarhus, not Aarhous

This is now corrected in the manuscript.

10. Lines 69, 302: Do not use the acronym CA. It is only mentioned one time after introducing it in the introduction, therefore it is not needed and clearer to write in full.

This is now corrected in the manuscript.

11. Line 310: add the word “brain” – brain regions is clearer

This is now corrected in the manuscript.

12. Line 373-4, blinding of the health care professionals: do you mean to the child to the exposure status of the child?

We have clarified this in the text: For example, it is not possible to blind a child’s exposure status from the health care professionals reviewing the child to rate the ND outcome

13. Line 414: add the word “potential” – associated with potential ND outcomes or potentially associated

This is now corrected in the manuscript.

14. Tables 2, 3: Not necessary to list every hospital name of the 10 from the Finnish study called PREDO.

We have deleted the hospital names

15. Tables: Be consistent when you write ICD-10 or ICD 10 (with or without hyphen)

We have now corrected this to ICD-10 throughout the manuscript

16. Table 5: Finland HILMO should include ICD-10 in recent years

We have now added this.

17. Table 5: Sweden ABIS study – should say Southeast or Southwest in geographic coverage. There is a mismatch. Also, “diagnosis codes” is vague. Is it ICD-10 like other sources?

The correct geographical region is South East, and the study uses ICD-9 or ICD-10 codes. This is now corrected in the manuscript.

18. Table 5: Sweden Clinical database for child and adolescent psychiatry in Sweden should say DSM-IV (not DSV-IV)

This is now corrected in the manuscript.

Reviewer #2: 1. The aim of this paper needs to be justified, given the project has been established.

2. The age of children would be important for the study of neurodevelopmental disorders. Thus, it would be very informative to list the start (end) year of the cohorts and registers to identify the children.

We agree with the comment that age at evaluation of development is very important, and many scales include an age range for use. The start/end dates of cohorts and registers would provide information about the generation rather than the age of subjects included. In some cases, the start year is included in the birth cohort name (such as Copenhagen Birth Cohort 2000). It would be a lot of work to include the start/end year for all data sources as some have closed enrolment, some have added waves of inclusion and others are still open. The purpose of our study was to identify those data sources with ND outcomes available, rather than the age of children included in these sources. 

3. L371-379: The authors stated that detection bias is always possible when using health registries. To my knowledge, detection bias is less likely in registers since the prescribers and clinicians who see the children are often different. Moreover, as the author mentioned, a large proportion of women take medications during pregnancy. Therefore, medications do not necessarily influence the doctors’ diagnosis. For instance, clinicians may be more reluctant to prescribe the medication to pregnant women unless they have severe disorders when an association is established. It is difficult to know whether any associations are due to positively biased practice or indication for treatment.

We acknowledge your comment and propose the following revision: “Finally, it should be considered that all data sources have potential for bias. In countries or regions with health registries, it may be comparatively cheap and fast to use diagnosis codes. However, detection bias cannot be fully ruled out. For example, it is not possible to blind a child’s exposure status from the health care professionals reviewing the child to rate the ND outcome. Whilst, this may be less of an influence early on, once an association between a medication exposure and a child ND outcome has been established this may positively bias practice, e. g. prescribing this medication only to clinically severe patients whose disease cannot be controlled with less toxic medications. Additional biases come from population health behaviour. For instance, it is suspected that women exposed to a suspected teratogen, or women with a history of mental illness, may be more likely to get health or developmental referrals for their children. “

4. There are still cohorts/registers missing from the ConcePTION project. Please clarify whether this project will be static, i.e., restricting to these 90 data sources, or be dynamic by identifying more relevant resources?

This study describes current electronic “linkable” data sources that can be used to assess neurodevelopmental outcomes in children. Other electronic data sources that have the potential to be linked to maternal and child outcomes will indeed be included in the catalogue developed within the CONCEPTION project. However, we cannot answer about the future updates and sustainability as these are not yet defined.

We hope that we have addressed all comments satisfactorily.

---

## [Decision Letter · Decision Letter 1]

27 Sep 2022

An inventory of European data sources to support pharmacoepidemiologic research on neurodevelopmental outcomes in children following medication exposure in pregnancy: A contribution from the ConcePTION project

PONE-D-21-33083R1

Dear Dr. Loane,

We’re pleased to inform you that your manuscript has been judged scientifically suitable for publication and will be formally accepted for publication once it meets all outstanding technical requirements.

Kind regards,

Maria Christine Magnus, PhD

Academic Editor

PLOS ONE

Reviewers' comments:

Reviewer's Responses to Questions

**Comments to the Author**

1. If the authors have adequately addressed your comments raised in a previous round of review and you feel that this manuscript is now acceptable for publication, you may indicate that here to bypass the “Comments to the Author” section, enter your conflict of interest statement in the “Confidential to Editor” section, and submit your "Accept" recommendation.

Reviewer #1: All comments have been addressed

Reviewer #2: All comments have been addressed

2. Is the manuscript technically sound, and do the data support the conclusions?

Reviewer #1: Yes

Reviewer #2: Yes

3. Has the statistical analysis been performed appropriately and rigorously? 

Reviewer #1: N/A

Reviewer #2: Yes

4. Have the authors made all data underlying the findings in their manuscript fully available?

Reviewer #1: Yes

Reviewer #2: Yes

5. Is the manuscript presented in an intelligible fashion and written in standard English?

Reviewer #1: Yes

Reviewer #2: Yes

6. Review Comments to the Author

Reviewer #1: (No Response)

Reviewer #2: (No Response)

7. PLOS authors have the option to publish the peer review history of their article (what does this mean?). If published, this will include your full peer review and any attached files.

Reviewer #1: No

Reviewer #2: No

---

## [Editor Report · Acceptance letter]

7 Oct 2022

PONE-D-21-33083R1 

An inventory of European data sources to support pharmacoepidemiologic research on neurodevelopmental outcomes in children following medication exposure in pregnancy: A contribution from the ConcePTION project 

Dear Dr. Loane:

I'm pleased to inform you that your manuscript has been deemed suitable for publication in PLOS ONE. Congratulations! Your manuscript is now with our production department. 

Kind regards, 

on behalf of

Dr. Maria Christine Magnus 

Academic Editor

PLOS ONE